# On the Low-Lying Electronically Excited States of Azobenzene Dimers: Transition Density Matrix Analysis

**DOI:** 10.3390/molecules26144245

**Published:** 2021-07-13

**Authors:** Evgenii Titov

**Affiliations:** Theoretical Chemistry, Institute of Chemistry, University of Potsdam, Karl-Liebknecht-Straße 24-25, 14476 Potsdam, Germany; titov@uni-potsdam.de

**Keywords:** azobenzene, dimer, transition density matrix, exciton, charge transfer, excited states, TD-DFT, ADC(2)

## Abstract

Azobenzene-containing molecules may associate with each other in systems such as self-assembled monolayers or micelles. The interaction between azobenzene units leads to a formation of exciton states in these molecular assemblies. Apart from local excitations of monomers, the electronic transitions to the exciton states may involve charge transfer excitations. Here, we perform quantum chemical calculations and apply transition density matrix analysis to quantify local and charge transfer contributions to the lowest electronic transitions in azobenzene dimers of various arrangements. We find that the transitions to the lowest exciton states of the considered dimers are dominated by local excitations, but charge transfer contributions become sizable for some of the lowest $\pi \pi^{*}$ electronic transitions in stacked and slip-stacked dimers at short intermolecular distances. In addition, we assess different ways to partition the transition density matrix between fragments. In particular, we find that the inclusion of the atomic orbital overlap has a pronounced effect on quantifying charge transfer contributions if a large basis set is used.

## 1. Introduction

Azobenzene is arguably the most famous molecular switch, which is widely employed in numerous applications at the nanoscale [1]. The key step promoting success of these applications is an isomerization reaction between *trans* and *cis* conformers, which is usually induced by light. The photochemistry of azobenzene has been a topic of numerous experimental and theoretical investigations [2]. Yet, in many systems employing azobenzene-containing molecules, azobenzene units may interact with each other, which, in turn, may affect their photophysical and photochemical properties. Association of azobenzene units occurs in self-assembled monolayers of azobenzene-containing molecules [3,4], in micelles of azobenzene-functionalized surfactants [5,6], and in aggregates formed upon surfactant complexation with polymers, such as DNA [7,8]. In addition, recent years have seen the emergence of covalently connected multiazobenzene molecules such as bisazobenzenes [9] and mixed dimers [10], star-shaped trimers [11,12,13] and tetramers [14], macrocycles [15] and molecular rings [16], tailored to applications ranging from energy storage [17] to wavelength-selective control of molecular switching [18]. Nanoscale architectures including azobenzene-functionalized carbon nanotubes [19] and metal-organic frameworks incorporating azobenzene [20] have also been realized.

The aggregation of molecular chromophores leads to a formation of collective electronic states, known as molecular excitons [21,22]. Classic theories of exciton formation, often used to interpret experimental findings, make use of monomer’s characteristics (wave function, excitation energy, and transition dipole moment) and dipole approximation for intermolecular interaction [21,22]. Specifically, in the classical Kasha exciton model [22], the excited state wave functions of a dimer are constructed from the excited (*e*) and the ground (*g*) state wave functions of monomers “1” and “2” as
(1)$$\psi_{\pm }=\frac{1}{\sqrt{2}}(\psi_{1}^{e}\psi_{2}^{g}\pm \psi_{1}^{g}\psi_{2}^{e})\;.$$
The excited state of the monomer (*e*) thus splits into two excited states (“±” linear combinations) in the dimer with corresponding exciton (Davydov) splitting of
(2)$$\Delta E=|E_{+}-E_{-}|\;.$$

In some cases, e.g., at short distances between monomers (when an overlap of molecular orbitals belonging to different chromophores is large), these theories should be amended by accounting for charge transfer excitations between monomers [23]. For aggregates composed of a relatively small number of chromophores it is possible to perform a full quantum chemical calculation instead of using exciton theories, thus avoiding approximations inherent to the latter [24,25]. (Naturally, the smallness of the system decreases with the increasing computational cost associated with a particular quantum chemical method.)

The nature of electronically excited states, obtained using quantum chemical calculations that treat a whole aggregate as a supermolecule, can be established by means of a natural transition orbital (NTO) analysis [26,27,28]. This analysis allows one to obtain a clearer picture of electronic transitions as compared with inspection of canonical molecular orbitals, especially in cases where multiple orbital pairs are involved in the transition in question, as is often the case for molecular complexes. In particular, this analysis, utilizing the singular value decomposition of an electronic single-particle transition density matrix (TDM), transforms the set of conventional molecular orbitals to the so-called “hole” and “particle” contributions, usually resulting in the reduction of dominant orbital pairs, describing the transition in question. We have used this approach to characterize the excited states and calculate the exciton splitting of free and adsorbed 4-nitroazobenzene dimers [29]. However, if NTOs are evenly delocalized over the aggregate they do not provide information on intrinsic structure of exciton states, i.e., they do not reveal the individual contributions of local (localized on a monomer) and charge transfer (occurring between monomers) excitations to a given electronic transition.

To decipher this, the TDM should be analyzed [27,30,31]. For the sake of clearness of further narration we present a theoretical description of TDM and its analysis below.

The reduced first-order spinless transition density matrix between a ground and an excited electronic state of a molecule (or a molecular complex) is defined as [32]
(3)$$\rho^{0I}\left(\vec{r},{\vec{r}}^{\;\prime };R\right)=N\;\int \int \;\cdots \int {\left[\Psi^{0}\left(\vec{x},{\vec{x}}_{2},\cdots ,{\vec{x}}_{N};R\right)\Psi^{I}\left({\vec{x}}^{\;\prime },{\vec{x}}_{2},\cdots ,{\vec{x}}_{N};R\right)\right]}_{\sigma^{\prime }\to \sigma }d{\vec{x}}_{2}\cdots d{\vec{x}}_{N}d\sigma \;.$$
Here, $\Psi^{0}\left({\vec{x}}_{1},{\vec{x}}_{2},\cdots ,{\vec{x}}_{N};R\right)$ stands for the *N*-electron ground state wave function and $\Psi^{I}\left({\vec{x}}_{1},{\vec{x}}_{2},\cdots ,{\vec{x}}_{N};R\right)$ stands for the *N*-electron excited state wave function of the *I*th electronically excited state. Both, $\Psi^{0}$ and $\Psi^{I}$, depend on 4N electronic variables, three spatial ($\vec{r}$) and one spin (σ) variable per an electron collected in a four-dimensional vector $\vec{x}$, and parametrically on the nuclear geometry R. In what follows we will not explicitly write the latter dependence for the sake of brevity, but this dependence should be kept in mind. We also assume all the quantities to be real.

In the case of configuration interaction singles (CIS), the ground state wave function is a single Slater determinant $\Phi_{0}$ constructed from occupied spin-orbitals χ. In this work we consider only closed-shell species and, therefore, write spin-orbitals χ as
(4)$$\chi_{2i-1}=\varphi_{i}\left(\vec{r}\;\right)\alpha \;,\;\chi_{2i}=\varphi_{i}\left(\vec{r}\;\right)\beta \;,\;i=1,\ldots ,N/2\;.$$
Here, φ are the spatial molecular orbitals (MOs), and α and β are spin-functions corresponding to spin-up and spin-down cases, respectively.

The ground state wave function thus reads:(5)$$\begin{array}{cc} \; & \Psi^{0}\left({\vec{x}}_{1},{\vec{x}}_{2},\cdots ,{\vec{x}}_{N}\right)=\Phi_{0}\left({\vec{x}}_{1},{\vec{x}}_{2},\cdots ,{\vec{x}}_{N}\right)=\\ \; & =\frac{1}{\sqrt{N!}}\left|\begin{array}{cccccccc} \varphi_{1}\alpha \left(1\right) & \varphi_{1}\beta \left(1\right) & \ldots & \varphi_{i}\alpha \left(1\right) & \varphi_{i}\beta \left(1\right) & \ldots & \varphi_{N/2}\alpha \left(1\right) & \varphi_{N/2}\beta \left(1\right)\\ \varphi_{1}\alpha \left(2\right) & \varphi_{1}\beta \left(2\right) & \ldots & \varphi_{i}\alpha \left(2\right) & \varphi_{i}\beta \left(2\right) & \ldots & \varphi_{N/2}\alpha \left(2\right) & \varphi_{N/2}\beta \left(2\right)\\ \vdots & \vdots & \ddots & \vdots & \vdots & \ddots & \vdots & \vdots \\ \varphi_{1}\alpha \left(N\right) & \varphi_{1}\beta \left(N\right) & \ldots & \varphi_{i}\alpha \left(N\right) & \varphi_{i}\beta \left(N\right) & \ldots & \varphi_{N/2}\alpha \left(N\right) & \varphi_{N/2}\beta \left(N\right) \end{array}\right|\;. \end{array}$$

The singlet excited state CIS wave function is a linear combination of singly excited Slater determinants $\Phi_{i\alpha }^{a\alpha }$ and $\Phi_{i\beta }^{a\beta }$, constructed from the $\Phi_{0}$ determinant substituting an occupied spin-orbital $\varphi_{i}\alpha$ by a virtual spin-orbital $\varphi_{a}\alpha$, or $\varphi_{i}\beta$ by $\varphi_{a}\beta$, respectively:(6)$$\Psi^{I}\left({\vec{x}}_{1},{\vec{x}}_{2},\cdots ,{\vec{x}}_{N}\right)=\sum_{i}\sum_{a}C_{ia}^{I}\left[\Phi_{i\alpha }^{a\alpha }\left({\vec{x}}_{1},{\vec{x}}_{2},\cdots ,{\vec{x}}_{N}\right)+\Phi_{i\beta }^{a\beta }\left({\vec{x}}_{1},{\vec{x}}_{2},\cdots ,{\vec{x}}_{N}\right)\right]\;,$$
where the $\Phi_{i\alpha }^{a\alpha }$ is
(7)$$\begin{array}{cc} \; & \Phi_{i\alpha }^{a\alpha }\left({\vec{x}}_{1},{\vec{x}}_{2},\cdots ,{\vec{x}}_{N}\right)=\\ \; & =\frac{1}{\sqrt{N!}}\left|\begin{array}{cccccccc} \varphi_{1}\alpha \left(1\right) & \varphi_{1}\beta \left(1\right) & \ldots & \varphi_{a}\alpha \left(1\right) & \varphi_{i}\beta \left(1\right) & \ldots & \varphi_{N/2}\alpha \left(1\right) & \varphi_{N/2}\beta \left(1\right)\\ \varphi_{1}\alpha \left(2\right) & \varphi_{1}\beta \left(2\right) & \ldots & \varphi_{a}\alpha \left(2\right) & \varphi_{i}\beta \left(2\right) & \ldots & \varphi_{N/2}\alpha \left(2\right) & \varphi_{N/2}\beta \left(2\right)\\ \vdots & \vdots & \ddots & \vdots & \vdots & \ddots & \vdots & \vdots \\ \varphi_{1}\alpha \left(N\right) & \varphi_{1}\beta \left(N\right) & \ldots & \varphi_{a}\alpha \left(N\right) & \varphi_{i}\beta \left(N\right) & \ldots & \varphi_{N/2}\alpha \left(N\right) & \varphi_{N/2}\beta \left(N\right) \end{array}\right|\;. \end{array}$$
The $\Phi_{i\beta }^{a\beta }$ is constructed analogously, substituting the $\varphi_{i}\beta$ column of $\Phi_{0}$ by the $\varphi_{a}\beta$ column. The expansion coefficients $C_{ia}^{I}$ obey the normalization condition (if orthonormal molecular orbitals are used, as is usually the case):(8)$$\sum_{i}\sum_{a}{\left(C_{ia}^{I}\right)}^{2}=\frac{1}{2}\;.$$

In the CIS case the TDM (Equation 3) takes the form of expansion in terms of occupied (*i*)-virtual (*a*) spatial orbital products:(9)$$\rho^{0I}\left(\vec{r},{\vec{r}}^{\;\prime }\right)=2\sum_{i}\sum_{a}C_{ia}^{I}\varphi_{i}\left(\vec{r}\right)\varphi_{a}\left({\vec{r}}^{\;\prime }\right)\;,$$
and one may define TDM in the MO basis, $P^{\left[\mathrm{MO}\right]}$:(10)$$P_{ia}^{\left[\mathrm{MO}\right]}=2C_{ia}^{I}\;.$$
The squared norm of the TDM is then:(11)$$\Omega =\;\int \int \;{\left[\rho^{0I}\left(\vec{r},{\vec{r}}^{\;\prime }\right)\right]}^{2}d\vec{r}d{\vec{r}}^{\;\prime }=2\;.$$

This result should not be confused with that for the squared norm of TDM with spin, which is equal to 1 [33] (see Appendix B for further details).

Further, expanding the spatial molecular orbitals in the basis of atomic orbitals (AOs) as
(12a)$$\begin{array}{cc} \; & \varphi_{i}\left(\vec{r}\right)=\sum_{\mu }c_{\mu i}\eta_{\mu }\left(\vec{r}\right)\;, \end{array}$$
(12b)$$\begin{array}{cc} \; & \varphi_{a}\left(\vec{r}\right)=\sum_{\nu }c_{\nu a}\eta_{\nu }\left(\vec{r}\right)\;, \end{array}$$
one may write the TDM as a linear combination of products of atomic orbitals:(13)$$\rho^{0I}\left(\vec{r},{\vec{r}}^{\;\prime }\right)=2\sum_{\mu }\sum_{\nu }\sum_{i}\sum_{a}C_{ia}^{I}c_{\mu i}c_{\nu a}\eta_{\mu }\left(\vec{r}\right)\eta_{\nu }\left({\vec{r}}^{\;\prime }\right)\;,$$
and define TDM in the AO basis, $P^{\left[\mathrm{AO}\right]}$ [34]:(14)$$P_{\mu \nu }^{\left[\mathrm{AO}\right]}=2\sum_{i}\sum_{a}C_{ia}^{I}c_{\mu i}c_{\nu a}\;.$$
The squared norm Ω then reads:(15)$$\Omega =\sum_{\mu }\sum_{\nu }{\left(P^{\left[\mathrm{AO}\right]}S\right)}_{\mu \nu }{\left({\mathrm{SP}}^{\left[\mathrm{AO}\right]}\right)}_{\mu \nu }=2\;,$$
where S is the AO overlap matrix:(16)$$S_{\mu \nu }=\int \eta_{\mu }\left(\vec{r}\right)\eta_{\nu }\left(\vec{r}\right)d\vec{r}\;.$$
This matrix can be calculated directly or from the MO coefficients matrix c [31] (with columns corresponding to different MOs):(17)$$S={\left(c^{T}\right)}^{-1}c^{-1}\;.$$

In the case of time-dependent density functional theory (TD-DFT) calculations, the so-called de-excitations should be taken into account when constructing the TDM [34,35]. Employing TD-DFT excited state “wave functions” in the CIS form (Equation 6) [36] with coefficients $C_{ia}^{I}$ being linear combinations of TD-DFT excitation $C_{i\to a}^{I}$ and de-excitation $C_{i\leftarrow a}^{I}$ coefficients, $C_{ia}^{I}=C_{i\to a}^{I}+C_{i\leftarrow a}^{I}$ [37,38,39], the $P^{\left[\mathrm{AO}\right]}$ takes the form:(18)$$P_{\mu \nu }^{\left[\mathrm{AO}\right]}=2\sum_{i}\sum_{a}C_{i\to a}^{I}c_{\mu i}c_{\nu a}+2\sum_{i}\sum_{a}C_{i\leftarrow a}^{I}c_{\mu i}c_{\nu a}\;.$$

In the case of wave function methods involving excitations higher than singles, one may construct approximate TDMs using only expansion coefficients corresponding to singly excited determinants if the electronic transitions are dominated by single-electron excitations.

The TDM in the AO basis can be further contracted to atoms or fragments to quantify how the excitation is distributed within a molecule or a complex [27,30,31]. This analysis allows one to determine the local (localized on fragments) and charge transfer (CT, occuring between fragments) contributions to the electronic transitions. Specifically, a “fraction of transition density matrix” (FTDM) matrix, with dimension of the number of fragments × the number of fragments, should be calculated. In this work, we compare five formulae to calculate the FTDM matrices:(19)$${}^{1}F_{XY}=\frac{\sum_{\mu \in X}\sum_{\nu \in Y}{\left(P_{\mu \nu }^{\left[\mathrm{AO}\right]}\right)}^{2}}{\sum_{\mu \in complex}\sum_{\nu \in complex}{\left(P_{\mu \nu }^{\left[\mathrm{AO}\right]}\right)}^{2}}\;,$$
(20)$${}^{2}F_{XY}=\frac{\sum_{\mu \in X}\sum_{\nu \in Y}\left|P_{\mu \nu }^{\left[\mathrm{AO}\right]}\right|}{\sum_{\mu \in complex}\sum_{\nu \in complex}\left|P_{\mu \nu }^{\left[\mathrm{AO}\right]}\right|}\;,$$
(21)$${}^{3}F_{XY}=\frac{\sum_{\mu \in X}\sum_{\nu \in Y}{\left(P^{\left[\mathrm{AO}\right]}S\right)}_{\mu \nu }{\left({\mathrm{SP}}^{\left[\mathrm{AO}\right]}\right)}_{\mu \nu }}{\sum_{\mu \in complex}\sum_{\nu \in complex}{\left(P^{\left[\mathrm{AO}\right]}S\right)}_{\mu \nu }{\left({\mathrm{SP}}^{\left[\mathrm{AO}\right]}\right)}_{\mu \nu }}\;,$$
(22)$${}^{4}F_{XY}=\frac{\frac{1}{2}\sum_{\mu \in X}\sum_{\nu \in Y}\left[{\left(P^{\left[\mathrm{AO}\right]}S\right)}_{\mu \nu }{\left({\mathrm{SP}}^{\left[\mathrm{AO}\right]}\right)}_{\mu \nu }+P_{\mu \nu }^{\left[\mathrm{AO}\right]}{\left({\mathrm{SP}}^{\left[\mathrm{AO}\right]}S\right)}_{\mu \nu }\right]}{\frac{1}{2}\sum_{\mu \in complex}\sum_{\nu \in complex}\left[{\left(P^{\left[\mathrm{AO}\right]}S\right)}_{\mu \nu }{\left({\mathrm{SP}}^{\left[\mathrm{AO}\right]}\right)}_{\mu \nu }+P_{\mu \nu }^{\left[\mathrm{AO}\right]}{\left({\mathrm{SP}}^{\left[\mathrm{AO}\right]}S\right)}_{\mu \nu }\right]}\;,$$
(23)$${}^{5}F_{XY}=\frac{\sum_{\mu \in X}\sum_{\nu \in Y}{\left(S^{1/2}P^{\left[\mathrm{AO}\right]}S^{1/2}\right)}_{\mu \nu }^{2}}{\sum_{\mu \in complex}\sum_{\nu \in complex}{\left(S^{1/2}P^{\left[\mathrm{AO}\right]}S^{1/2}\right)}_{\mu \nu }^{2}}\;.$$
Here, *X* and *Y* denote molecular fragments, e.g., monomers of a dimer. The ${}^{1}F$ and ${}^{2}F$ matrices are calculated employing $P^{\left[\mathrm{AO}\right]}$ only. The ${}^{1}F$ matrix is often employed in the context of semiempirical calculations [40,41]. The ${}^{2}F$ matrix is another way to empirically partition the $P^{\left[\mathrm{AO}\right]}$ matrix, which is one of the options in the Multiwfn program [42]. We note, however, that there is no rigorous justification for Equations (Equation 19) and (Equation 20).

The ${}^{3}F$, ${}^{4}F$, and ${}^{5}F$ matrices include, in addition, AO overlap matrix S. They correspond to the so-called charge-transfer numbers [33] (see also [43,44]):(24)$$\Omega_{XY}=\int_{X}\int_{Y}{\left[\rho^{0I}\left(\vec{r},{\vec{r}}^{\;\prime }\right)\right]}^{2}d\vec{r}d{\vec{r}}^{\;\prime }\;,$$
which represent the partitioning of Ω (Equation 11) between fragments. Formulae (Equation 21)  [31] and (Equation 22) [33] are similar to Mulliken population analysis. Formula (Equation 23) [45,46] is similar to Löwdin population analysis. Formulae similar to Equations (Equation 21)–(Equation 23) are used in the TheoDORE package [47]. We note that the sum of all FTDM elements, $\sum_{XY}{}^{i}F_{XY}\;,\;i\;=\;1\ldots 5$, is one for our definitions (Equation 19)–(Equation 23). Therefore, we will sometimes express the FTDM elements in %, i.e., as ${}^{i}F_{XY}\times 100$.

In this work, we apply the TDM analysis (Equations (Equation 19)–(Equation 23)) to azobenzene dimers in order to characterize local and charge transfer excitations participating in the lowest electronic transitions. Moreover, we compare the results obtained with different FTDM formulae.

## 2. Results

### 2.1. Monomer

We first briefly discuss excited states of an azobenzene monomer. While the excited states of azobenzene were widely investigated before [48,49,50,51], we concentrate here on the performance of methods which will be used for dimers, and on the transition density matrix (TDM) analysis of the monomeric states. In particular, we will use linear response time-dependent density functional theory (TD-DFT) [36] with the B3LYP [52,53] and ωB97X-D [54] functionals, and algebraic diagramatic construction through second order (ADC(2)) [55,56,57]. The B3LYP is a global hybrid functional (with 20% of exact exchange) and the ωB97X-D is a long-range corrected functional. They are known to yield different excitation energies for azobenzenes (the TD-B3LYP energies are more red-shifted) and perform differently for charge transfer transitions [29,58,59]. The ADC(2) method is a post-Hartree–Fock wave function-based method, which was found to perform similarly to the approximate second-order coupled cluster (CC2) [60] for azobenzene in monomeric and dimeric forms (see supporting information in [61]).

In Table 1 we list excitation energies and oscillator strenghts of the lowest five transitions of *trans* azobenzene calculated with TD-B3LYP and TD-ωB97X-D in combination with three basis sets of increasing size [6-31G* (double zeta basis with polarization functions on non-hydrogen atoms) [62,63], def2-TZVP (triple zeta basis with polarization functions on all atoms) [64], and aug-cc-pVTZ (triple zeta basis with polarization and diffuse functions on all atoms) [65,66] for two geometries, optimized with B3LYP/def2-TZVP and B3LYP/6-31G*, respectively. The same information but for ADC(2) calculations with five different basis sets (three correlation consistent basis sets from double to quadruple zeta quality augumented with diffuse functions [65,66], and two Karlsruhe basis sets of triple and quadruple zeta quality [64]) are shown in Table 2. The dominant natural transition orbitals (NTOs) of these five transitions calculated at three selected levels, TD-B3LYP/def2-TZVP, TD-ωB97X-D/def2-TZVP, and ADC(2)/aug-cc-pVTZ, are shown in Figure 1.

The $S_{1}$ state has $n\pi^{*}$ character and states $S_{2}$–$S_{5}$ are $\pi \pi^{*}$ states, for all considered methods. Excitation energies increase in the order TD-B3LYP < TD-ωB97X-D < ADC(2), when the same basis set and the same geometry are considered. The $S_{0}\to S_{1}$ excitation energy is virtually independent of the basis set for TD-DFT methods and differences of not larger than only 0.04 eV are observed for ADC(2) when changing the basis set (while keeping the geometry). The basis set dependence is more pronounced for the $S_{2}$ state. For the TD-DFT methods, the $S_{0}\to S_{2}$ excitation energy drops by 0.11–0.12 eV when going from the 6-31G* basis to the def2-TZVP basis, and further by 0.02–0.03 eV when changing to the aug-cc-pVTZ basis. In the case of ADC(2), the variation in the $S_{0}\to S_{2}$ excitation energy with basis set is within 0.06 eV, for considered basis sets. We note that the ADC(2) excitation energies obtained with the aug-cc-pVDZ basis set are similar to those calculated with the def2-TZVP basis, and aug-cc-pVTZ energies are virtually the same as the def2-QZVP energies. The aug-cc-pVQZ energies are lower than the aug-cc-pVTZ energies by no more than 0.02 eV.

Further, the excitation energies exhibit geometry dependence as also reported in Ref. [51]. The $S_{0}\to S_{1}$ excitation energies are higher by 0.02–0.03 eV for the B3LYP/6-31G* geometry than for the B3LYP/def2-TZVP geometry. The $S_{0}\to S_{2}$ excitation energies are, on the contrary, lower by 0.08–0.09 eV for the B3LYP/6-31G* geometry than for the B3LYP/def2-TZVP geometry.

The $S_{3}$ and $S_{4}$ states are nearly degenerate with the dominant hole NTO located on phenyl rings and the dominant particle NTO being very similar to the dominant particle NTO of the $S_{0}\to S_{2}$ transition. All considered methods except TD-B3LYP/6-31G* predict a higher oscillator strength (0.03–0.05) for the $S_{0}\to S_{3}$ transition than for the $S_{0}\to S_{4}$ transition (0.00). The $S_{5}$ state is a dark $\pi \pi^{*}$ state separated by ∼0.7 eV from the $S_{3}$ and $S_{4}$ states. Basis set and geometry dependencies of the excitation energies of these $\pi \pi^{*}$ states are similar to those for $S_{2}$ (see above).

In the next section of this contribution we will characterize dimeric excited states by means of a transition density matrix analysis (described in detail in Section 1). It is therefore interesting to see how this analysis works for the monomer, and how different the results obtained using different “fraction of transition density matrix” (FTDM) matrices (Equations (Equation 19)–(Equation 23)) are. To this aim we divide the azobenzene molecule into three fragments, two phenyl fragments C${}_{6}$H${}_{5}$ (termed Ph${}_{1}$ and Ph${}_{2}$) and the azo fragment (termed NN). In Figure 2 and Figure 3 we report FTDM matrices ${}^{1}F,\ldots {,}^{5}F$ (each of the 3 × 3 size) for the monomeric $S_{0}\to S_{1}$ and $S_{0}\to S_{2}$ transitions, respectively.

Looking first at the ${}^{3}F{,}^{4}F$, and ${}^{5}F$ matrices (which are more rigorous than the ${}^{1}F$ and ${}^{2}F$ matrices) of Figure 2 we see that the largest contribution to the $S_{0}\to S_{1}$ transition is a local excitation of the azo group (the 2,2 element), and the second largest contributions come from charge transfer excitations from the azo group to the phenyl groups (the 2,1 and 2,3 elements). This picture correlates well with what may be expected when inspecting the hole and particle NTOs of the $S_{0}\to S_{1}$ transition, shown in Figure 1, left.

Further, the ${}^{3}F$ and ${}^{4}F$ matrices are almost identical to each other, and the ${}^{5}F$ matrix, while qualitatively being the same, is quantitatively different from ${}^{3}F$ and ${}^{4}F$. In particular, the local NN contribution is smaller for ${}^{5}F$ than for ${}^{3}F$ and ${}^{4}F$ (by up to ∼24%, see the rightmost column of Figure 2). The ${}^{1}F$ matrices are qualitatively similar to the ${}^{3}F,\ldots {,}^{5}F$ matrices only for TD-DFT calculations with 6-31G* and aug-cc-pVTZ basis sets, and the ${}^{2}F$ matrices differ considerably from the ${}^{3}F,\ldots {,}^{5}F$ matrices.

For the $S_{0}\to S_{2}$ transition, the TDM analysis reveals that the largest contributions are Ph → NN charge transfer excitations and the second largest contributions are the local excitations of phenyls. Interestingly, the Ph${}_{1}$→ Ph${}_{2}$ and Ph${}_{2}$→ Ph${}_{1}$ “long-range” CT contributions are also rather strong.

### 2.2. Dimers

In this work, we will consider the dimers shown in Figure 4. The studied dimeric arrangements include a cofacial π-stacked dimer (Figure 4a), a slip-stacked dimer (Figure 4b), and an in-plane dimer (Figure 4c). Moreover, we will consider an optimized dimer model, which will be discussed below (in Section 2.2.5).

#### 2.2.1. Cofacial π-Stacked Dimer d=3.5 Å

We first discuss a cofacial π-stacked dimer with a distance between monomers of 3.5 Å (see Figure 4a, d=3.5 Å). It was used as a model system to study dimerization effects on photoinduced dynamics in our previous work [61]. The excitation energies and oscillator strengths calculated with the two TD-DFT methods (TD-B3LYP and TD-ωB97X-D) and the ADC(2) method (all at the geometry constructed from the B3LYP/def2-TZVP monomer geometry) are listed in Table 3. We see that the three methods predict different spectra, in particular the bright transition is $S_{0}\to S_{9}$ at the TD-B3LYP level, $S_{0}\to S_{4}$ at the TD-ωB97X-D level, and $S_{0}\to S_{6}$ at the ADC(2) level. At the same time, all three methods predict a blue shift of the bright transition in comparison to the monomer, as expected based on the Kasha theory [22] (compare Table 1, Table 2 and Table 3). This shift is 0.14 eV for TD-DFT and 0.05 eV for ADC(2). To make a connection to experimental studies, we note that blue shifts of about 0.1 eV have been observed upon micellization of azobenzene-functionalized surfactants [5], and blue shifts of about 0.6 eV have been found in azobenzene-containing self-assembled monolayers [3].

The dominant NTOs for the ten lowest transitions are shown in Figure 5. First of all, it is seen that each NTO is delocalized over dimer. Therefore, it is impossible to judge either the transitions are composed of local excitations, charge-transfer excitations, or a mixture of both. Secondly, one can correlate dimeric NTOs with the monomeric ones. For example, at the TD-B3LYP level, the NTOs of transitions $S_{0}\to S_{1}$, $S_{0}\to S_{2}$, $S_{0}\to S_{4}$, and $S_{0}\to S_{5}$ look similar to the NTOs of the $S_{0}\to S_{1}$ monomeric transition, the NTOs of transitions $S_{0}\to S_{3}$, $S_{0}\to S_{6}$, and $S_{0}\to S_{9}$ correspond to the NTOs of the $S_{0}\to S_{2}$ monomeric transition, etc. Expectedly, the NTOs of the bright dimeric transitions match the $S_{0}\to S_{2}$ monomeric NTOs.

It is known that density functional approximations with a relatively low fraction of exact exchange, e.g., B3LYP, which includes 20% of exact exchange, promote CT excitations to a lower part of spectrum [67,68,69]. In our previous publication we observed it for a slightly asymmetric 4-nitroazobenzene dimer model (constructed using results of an optimizaton of a dimer attached to a silicon cluster), applying the NTO analysis [29]. However, for the symmetric model considered here, a visual inspection of NTOs cannot provide information about intrinsic composition of the electronic transitions, as mentioned above. Contrary to the NTOs, FTDM matrices are devised to answer the question of what types of excitations contribute to a transition. The ${}^{5}F$ matrices for the ten lowest transitions of the dimer d=3.5 Å are shown in Figure 6.

At the TD-B3LYP level, while the first two transitions are predominantly composed of two local excitations (large diagonal elements of FTDM), the transitions $S_{0}\to S_{3},\ldots ,S_{0}\to S_{8}$ are either dominated by CT excitations (large off-diagonal elements of FTDM) or represent an even mixture of local and CT excitations (∼1:1:1:1 ratio of four FTDM elements). The delocalized dimeric transitions dominated by local excitations are commonly termed Frenkel excitons and those dominated by CT excitatios are termed charge-resonance transitions [31]. The bright transition $S_{0}\to S_{9}$ is a Frenkel exciton.

At the TD-ωB97X-D level and the ADC(2) level the transitions up to the bright one are dominated by local excitations. However, considering a splitting of the $S_{0}\to S_{2}$ monomeric transition we see that the CT contributions are much larger for the dark, lower energy transition than for the bright transition. Namely, while the bright transition ($S_{0}\to S_{4}$ at the TD-ωB97X-D level and $S_{0}\to S_{6}$ at the ADC(2) level) may be clearly assigned as a Frenkel exciton, the corresponding dark, lower energy transition ($S_{0}\to S_{3}$ at both levels of theory) involves rather large CT contributions (off-diagonal elements of ∼14% for TD-ωB97X-D and ∼16% for ADC(2)). Therefore, the formation of the exciton states from the bright monomeric transition goes beyond the Kasha model that is based solely on locally excited states as will be discussed in Section 3.

It is also interesing to inspect how different FTDM formulae (Equations (Equation 19)–(Equation 23)) work for the dimer. In Figure 7 we show the ${}^{1}F,\ldots {,}^{5}F$ matrices for the $S_{0}\to S_{1}$ transition calculated at four levels of theory (TD-ωB97X-D/6-31G*, TD-ωB97X-D/def2-TZVP, TD-ωB97X-D/aug-cc-pVTZ, and ADC(2)/aug-cc-pVTZ) using dimer’s partitioning to monomers. (The excitation energies and oscillator strengths calculated using these four methods are shown in the Appendix A.)

Compared to ${}^{3}F,\ldots {,}^{5}F$, the ${}^{1}F$ matrix gives a similar result if the 6-31G* or def2-TZVP basis sets are used. However, with the aug-cc-pVTZ basis set CT elements of ${}^{1}F$ become much larger and deviation from ${}^{3}F,\ldots {,}^{5}F$ is sizable. The ${}^{2}F$ matrices show appreciable CT contributions already with a small basis set (∼10% with 6-31G*) and thus differ from the other matrices, similarly to the monomeric case (compare with Figure 2, “6-31G*” columns). In the case of ADC(2)/aug-cc-pVTZ, the ${}^{1}F$ and ${}^{2}F$ matrices differ considerably from ${}^{3}F,\ldots {,}^{5}F$ (see the rightmost column of Figure 7). While the former predict an even mixture of local and CT excitations, the latter yield a Frenkel exciton. We also note small negative off-diagonal elements for the ${}^{3}F$ and ${}^{4}F$ matrices together with the corresponding diagonal elements being slightly larger than 50%. This result can be explained by the use of dominant expansion coefficients only (a default TURBOMOLE printing level was used). This demonstrates that ${}^{3}F_{XY}$ and ${}^{4}F_{XY}$ may, in principle, become negative. Conversely, the ${}^{5}F_{XY}$ elements are always positive by construction of the ${}^{5}F$ matrix (see Equation (Equation 23)).

Further, if the CT contributions are non-negligible, as is the case for the $S_{0}\to S_{3}$ transition, one may wonder from which part of one monomer to which part of the other monomer an electron is transferred. To answer this question, we partitioned the dimer into phenyl and azo fragments, similarly to what was done for the monomer. This partitioning is shown in Figure 8a. The corresponding 6 × 6 ${}^{5}F$ matrices calculated with TD-ωB97X-D/def2-TZVP and ADC(2)/aug-cc-pVTZ are presented in Figure 8b and c, respectively.

It is seen that the largest CT contributions between monomers are from phenyl groups to the azo groups (FTDM elements 1,5; 3,5; 4,2; 6,2, which amount to 2.5% each). The next largest CT elements at the TD-ωB97X-D level are CTs between neighbouring phenyls, i.e., Ph${}_{1}↔$ Ph${}_{3}$ and Ph${}_{2}↔$ Ph${}_{4}$, which are 2.1% each. The “cross”-contributions Ph${}_{1}↔$ Ph${}_{4}$ and Ph${}_{2}↔$ Ph${}_{3}$ are 1.3% at the TD-DFT level. On the contrary, the ADC(2) calculation results in equal neighbouring and “cross” CT terms, which amount to 2.2% each at the ADC(2) level. However, it may be a result of truncation of a vector with expansion coefficients.

#### 2.2.2. Stacked Arrangement

Further, we investigated the effect of intermolecular separation on the lowest excited states using the TD-ωB97X-D/def2-TZVP level of theory. The intermolecular distance *d* between molecular planes (see Figure 4a) was varied from 3.0 to 10.0 Å with a step of 0.5 Å. The excitation energy spectra as a function of distance *d* are shown in Figure 9a. The bars are colored according to the oscillator strength of the transitions, with a darker color corresponding to a larger oscillator strength. The bright state labels are printed.

At the TD-ωB97X-D/def2-TZVP level of theory, the bright transition is $S_{0}\to S_{4}$ for all *d* values except d=3.0 Å. For the latter, the bright transition is $S_{0}\to S_{7}$. The NTOs and FTDMs for dimer d=3.0 Å are shown in the SI (Appendix A). The $S_{1}$ and $S_{2}$ states rapidly become nearly degenerate with increasing intermolecular distance. The zoomed-in distance dependence of $S_{3}$ and $S_{4}$ excitation energies is shown in Figure 9b. Expectedly, the $S_{3}$–$S_{4}$ splitting reduces with distance. Yet, the $S_{3}$ excitation energy rises faster than the $S_{4}$ excitation energy drops. We also show a center relative to which the $S_{3}$ and $S_{4}$ states split, i.e., $\frac{E_{S_{3}}+E_{S_{4}}}{2}$ (black crosses in Figure 9b). It is seen that the center lies below the monomeric $S_{2}$ excitation energy (shown with a light grey line) at shorter distances, and approaches it with increasing separation. The oscillator strength of the dimer’s bright transition grows with increasing intermolecular distance (see inset in Figure 9b), similarly to what has been observed at the TD-B3LYP level for dimers of CF${}_{3}$-Azo-OCH${}_{3}$ molecules [4]. However, it remains smaller than twice oscillator strength of the monomer (of the $S_{0}\to S_{2}$ monomeric transition) for all considered intermolecular distances (up to 10 Å). We recall that from Kasha’s exciton model [22] one expects enhancement of the transition dipole moment in $\sqrt{2}$ times, and, hence, enhancement of the oscillator strength in $2\frac{E_{dimer}^{S_{4}}}{E_{monomer}^{S_{2}}}$ times, in comparison to the corresponding values for the $S_{0}\to S_{2}$ transition of the monomer. (We note that the oscillator strength *f* is determined by transition dipole moment μ and excitation energy *E* as $f=\frac{2m_{e}E\mu^{2}}{3\hbar^{2}e^{2}}$ [70]. Here, *ℏ* is the reduced Planck constant, $m_{e}$ and *e* are the mass and the charge of the electron, respectively.)

In Figure 9c we show the diagonal (1,1) and off-diagonal (1,2) elements of the 2×2 FTDM matrices ${}^{5}F$ for the lowest four transitions ($S_{0}\to S_{1},\ldots ,S_{0}\to S_{4}$) as a function of distance *d*. We see that the $S_{0}\to S_{3}$ transition possesses a non-negligible CT contribution at d<5.0 Å, which decays with distance and becomes virtually zero at d>5.0 Å. The $S_{4}$ state, on the contrary, is largely dominated by the local excitations for all considered distances except d=3 Å, at which the $S_{4}$ state has a different character and does not correspond to the $S_{2}$ monomeric state (see NTOs in Appendix A). The $S_{0}\to S_{1}$ and $S_{0}\to S_{2}$ transitions show appreciable CT contributions (>1%) only at d=3.0 Å and d=3.5 Å. The $S_{1}$ CT contributions are slightly larger than the $S_{2}$ ones.

#### 2.2.3. Slip-Stacked Arrangement

Next, we studied a slip-stacked arrangement with the second molecule sliding in its plane as shown in Figure 4b. The distance between molecular planes was kept at 3.5 Å. The distance dependence of excitation energies and FTDM matrix elements is shown in Figure 10. The bright state is $S_{4}$ up to d=5.5 Å and changes to $S_{3}$ at larger distances. The center relative to which the $S_{3}$ and $S_{4}$ states split lies below the monomeric $S_{2}$ energy. The excitation energies and FTDM matrix elements of $S_{0}\to S_{3}$ and $S_{0}\to S_{4}$ transitions, as well as the oscillator strength of the bright transition, show non-monotonic *d*-dependence. The CT contribution to the bright transition ($S_{0}\to S_{4}$) reaches a maximum (∼8%) at sliding coordinate of 2 Å. The CT contribution to the $S_{0}\to S_{3}$ transition is largest at d=0 and amounts to ∼14%. We also note that at d=4.5 Å FTDM elements for $S_{1}$ and $S_{2}$ states show unexpected behaviour: while off-diagonal elements ${}^{5}F_{12}$ are almost the same, the diagonal elements ${}^{5}F_{11}$ differ by ∼0.013. That means that the sum ${}^{5}F_{11}{+}^{5}F_{12}$ is smaller than 0.5 for the $S_{0}\to S_{2}$ transition, at d=4.5 Å. This counterintuitive result can be explained by degeneracy of the $S_{1}$ state ($E_{S_{1}}=E_{S_{2}}=2.6450$ eV at d=4.5 Å), leading to non-symmetric FTDM matrices.

#### 2.2.4. In-Plane Arrangement

We have also considered an in-plane arrangement with two azobenzene molecules lying in the same plane as shown in Figure 4c. The results are presented in Figure 11. The $S_{4}$ state is the bright state for all considered distances (from 5.5 to 10.0 Å). The CT contributions are small for all in-plane dimers. The largest CT contribution of only ∼0.7% (the off-diagonal element ${}^{5}F_{12}$) is found for the $S_{0}\to S_{3}$ transition of the dimer d=5.5 Å. The lowest states are therefore nearly pure Frenkel excitons.

#### 2.2.5. Optimized Dimer

Finally, we optimized the geometry of azobenzene dimer starting from the cofacial π-stacked dimer d=3.5 Å using B3LYP with Grimme’s D3 correction including the Becke–Johnson (BJ) damping function [71]. The optimized geometry is shown in Figure 12. It is seen that there are longitudinal and transverse shifts of one molecule with respect to the other one. The occurrence of these shifts is reminiscent to the case of the benzene dimer, for which the slip-stacked geometry is energetically favored [72]. The shortest intermolecular CN distances are about 3.46 Å. Moreover, the phenyl rings are slightly rotated around NC bonds with the corresponding NNCC dihedral angles being in the range 2–5${}^{\circ }$.

The excitation energies and oscillator strengths of the lowest ten transitions of the optimized dimer calculated with TD-B3LYP/def2-TZVP, TD-ωB97X-D/def2-TZVP, and ADC(2)/aug-cc-pVTZ are listed in Table 4. The bright transition appears at 4.10 eV (4.09 eV) at the TD-ωB97X-D (ADC(2)) level of theory, i.e., 0.11 eV lower than the corresponding transition in the cofacial π-stacked dimer d=3.5 Å (compare with Table 3). At the ADC(2) level, the bright transition is now $S_{0}\to S_{4}$ and not $S_{0}\to S_{6}$ as for the cofacial dimer. The brightest transition at the TD-B3LYP level is $S_{0}\to S_{10}$, which appears at 3.84 eV. There are also several transitions bearing small oscillator strengths below $S_{0}\to S_{10}$. The dominant NTOs for the ten lowest transitions of the optimized dimer are shown in Figure 13. Again, they are evenly delocalized over the dimer.

The FTDM ${}^{5}F$ matrices are shown in Figure 14. It is seen that the $S_{0}\to S_{4}$ transition obtained at the TD-ωB97X-D and ADC(2) levels possesses larger CT contributions than the $S_{0}\to S_{3}$ transition, in contrast to the π-stacked dimer d=3.5. In other words, the bright transition shows stronger CT than the dark transition does. The $S_{3}$ and $S_{4}$ states calculated with TD-B3LYP are charge-resonance $n\pi^{*}$ states. Moreover, the FTDM matrices of the transitions to these states are non-symmetric, owing again to the degeneracy of the $S_{3}$ ($S_{4}$) state.

## 3. Discussion

Based on the Kasha theory [22], one expects a formation of two states out of each monomeric excited state (see Equation (Equation 1)). In particular, two $n\pi^{*}$ states corresponding to the $S_{1}$ state of azobenzene and two $\pi \pi^{*}$ states corresponding to the $S_{2}$ state of azobenzene are expected in the low energy region of the azobenzene dimer spectrum. In the simplest case, they are anticipated to occur as the first four excited states $S_{1}$–$S_{4}$ of the dimer. Indeed, the calculations with the long-range corrected funtional ωB97X-D show that the two lowest dimeric excited states originate from the $S_{1}$ ($n\pi^{*}$) monomeric state and the next two from the $S_{2}$ ($\pi \pi^{*}$) state for all the considered dimers except the π-stacked dimer d=3.0 Å. This simplest expectation is also realized at the ADC(2) level for the optimized dimer, whereas for the π-stacked dimer d=3.5 the $S_{3}$ and $S_{6}$ states correspond to the monomeric $S_{2}$ state, and the $S_{4}$ and $S_{5}$ dimeric states correspond to the $S_{3}$/$S_{4}$ monomeric states.

Yet, the Kasha exciton model does not include CT excitations between monomers, as can be seen from Equation (Equation 1). The simplest way to think of them is to imagine a dimer with two MOs per monomer, one occupied (*H*) and one virtual (*L*) MO (*H* stands for HOMO, highest occupied molecular orbital, and *L* stands for LUMO, lowest unoccupied molecular orbital) [73,74,75,76]. (For analysis including more than two monomer orbitals see [77].) In the ground state, the $H_{1}$ and $H_{2}$ orbitals are each doubly occupied. The excited state of the monomer is the H→L excitation. The excited states of the dimer are constructed as linear combinations of $H_{1}\to L_{1}$, $H_{2}\to L_{2}$, $H_{1}\to L_{2}$, and $H_{2}\to L_{1}$ excitations. The $H_{1}\to L_{1}$ and $H_{2}\to L_{2}$ are the local excitations (accounted in the Kasha model), and the $H_{1}\to L_{2}$ and $H_{2}\to L_{1}$ are the intermolecular CT excitations (not present in the Kasha model). Therefore, one expects a formation of four excited states in the dimer out of one excited state of the monomer [76].

Returning to the simplest scenario for the azobenzene dimer, with the $S_{1}$ and $S_{2}$ states resulting from the $S_{1}$ state of the monomer and $S_{3}$ and $S_{4}$ states resulting from the $S_{2}$ state of the monomer, which fits into the Kasha model, it is a priori not possible to exclude participation of the CT excitations in the formation of the dimeric excited states. The delocalized NTOs of the symmetric dimer (i.e., the dimer with the geometrically and compositionally identical monomers) do not provide an answer as to whether the transition is composed of the local or CT excitations. The answer instead comes from the TDM analysis described in Section 1.

We find that one of the lowest dimeric states corresponding to the $S_{2}$ state of the monomer involves substantial CT excitations at short intermolecular distances. It is the $S_{4}$, bright state of the optimized dimer (characterized by the off-digonal element of FTDM of ∼8% at the TD-ωB97X-D level and ∼9% at the ADC(2) level) and the $S_{3}$, dark state of the cofacial π-stacked dimer d=3.5 Å (the off-diagonal FTDM element of ∼14% at the TD-ωB97X-D level and ∼16% at the ADC(2) level). The CT contributions to the lowest dimeric excited states decay with increasing intermolecular distance. Varying the sliding coordinate results in non-monotonic CT dependencies for the $S_{3}$ and $S_{4}$ states, showing larger CT contributions for the $S_{3}$, dark state (than for the $S_{4}$, bright state) in the π-stacked dimer (at sliding coordinate ≈0) and, oppositely, larger CT contributions for the $S_{4}$ state at values of the sliding coordinate close to 2 Å as well as for the optimized dimer. For the considered in-plane dimers the CT contributions are negligible.

Regarding different ways to analyze TDM (Equations (Equation 19)–(Equation 23)), the inclusion of the atomic orbital overlap matrix S has a pronounced effect on FTDM matrix elements if a large basis set is used. On the other hand, an approximate Formula (Equation 19) provides results very similar to those obtained with more complex Formulae (Equation 21)–(Equation 23) if a small basis set (6-31G* in our case) is employed (see Figure 7). Therefore, the use of this formula (Equation (Equation 19)) can be justified for analysis of local and charge transfer contributions to the excited states of noncovalent dimers and larger aggregates in the case of quantum chemical calculations employing rather small basis sets  [25,78]. We also note that FTDM matrices including the AO overlap (Equations (Equation 21)–(Equation 23)) are basis set dependent, similarly to Mulliken and Löwdin population analyses. And, moreover, the Mulliken-type FTDM matrices (Equations (Equation 21) and (Equation 22)) may, in principle, have unphysical negative elements (“fractions”). Finally, it should be noted that there are other approaches for an atom-related analysis of electronically excited states [79,80].

Future research will be devoted to the exploration of exciton states of larger azobenzene aggregates by means of quantum chemical calculations and transition density matrix analysis as well as to the elucidation of exciton dynamics in these systems.

## 4. Materials and Methods

The B3LYP functional was used for all geometry optimizations. The geometry of the azobenzene monomer was optimized in the electronic ground state using the def2-TZVP basis set. In addition, the geometry was reoptimized using a smaller basis, 6-31G*, to analyze the geometry effect on excitation energies.

Dimer models were constructed from the def2-TZVP optimized monomer geometry by translating this geometry in a desired direction. The stacked, slip-stacked, and in-plane arrangements were studied. Furthermore, the dimer geometry was optimized at the B3LYP+D3(BJ)/def2-TZVP level of theory. The geometries in the .xyz format are available in the SI.

Excitation energies and oscillator strengths of the lowest electronic transitions were computed using TD-DFT (with B3LYP and ωB97X-D functionals) and ADC(2) methods with various basis sets as specified in Section 2. Twenty and ten excited states were requested in TD-DFT and ADC(2) calculations, respectively.

DFT and TD-DFT calculations were performed using Gaussian 16 [81], and the ADC(2) calculations were done using TURBOMOLE V7.0 [82,83]. The ADC(2) calculations employed the resolution-of-the-identiy approximation [84,85,86] together with the optimized auxiliary basis sets [87,88,89], and the frozen core approximation as implemented in TURBOMOLE.

The electronic transitions were analyzed using transition density matrices (TDMs) and natural transition orbitals (NTOs). TDMs were calculated with Multiwfn 3.7 [42] in the case of Gaussian calculations and constructed from wave function expansion coefficients and molecular orbital (MO) coefficients in the case of TURBOMOLE calculations. NTOs were calculated using Multiwfn 3.7 in the case of Gaussian calculations and with TURBMOLE’s ricctools in the case of the TURBOMOLE calculations.

## 5. Conclusions

We applied transition density matrix (TDM) analysis to characterize low-lying excited states of the azobenzene monomer and dimers. The excited states were calculated using TD-DFT and ADC(2) methods. Various dimeric arrangements, including co-facial π-stacked, slip-stacked, in-plane, as well as an optimized dimer were considered. The TDM analysis revealed that local, monomeric excitations prevail over charge transfer (CT), inter-monomeric excitations in the composition of the low-lying dimeric, exciton states. The CT contributions, however, are pronounced for some of the $\pi \pi^{*}$ states of the stacked and slip-stacked dimers at short intermolecular distances. Furthermore, we compared different ways to partition the TDM (see Equations (Equation 19)–(Equation 23)). We found that omission of an atomic orbital overlap matrix has a strong effect on the “fraction of transition density” (FTDM) matrices if a large basis set is used, but leads to a minor change in case of a moderate basis set.

## Figures and Tables

**Figure 1 molecules-26-04245-f001:**
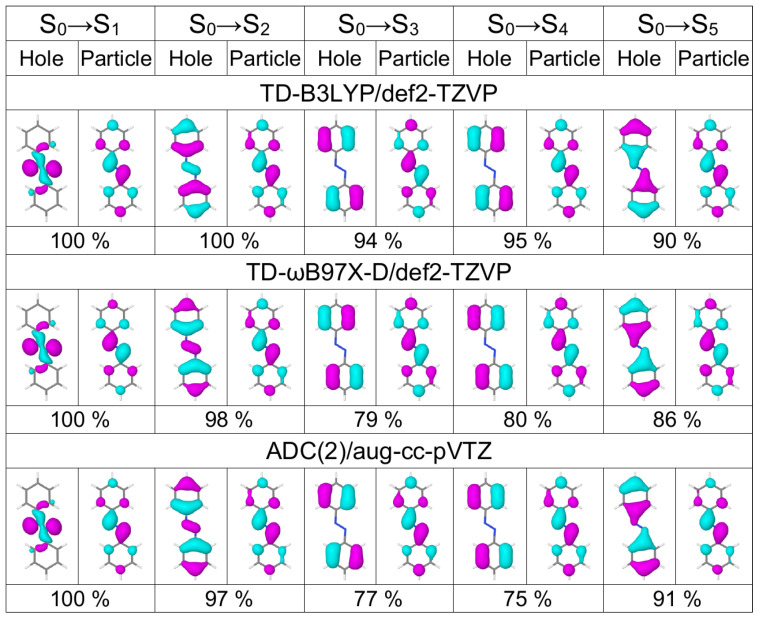
Dominant natural transiion orbital pairs for the lowest five transitions of the azobenezene monomer calculated at the TD-B3LYP/def2-TZVP (**top row**), TD-ωB97X-D/def2-TZVP (**middle row**), and ADC(2)/aug-cc-pVTZ (**bottom row**) levels of theory. Percentage shows the contribution of a given hole–particle pair to a transition. The monomer geometry is optimized at the B3LYP/def2-TZVP level.

**Figure 2 molecules-26-04245-f002:**
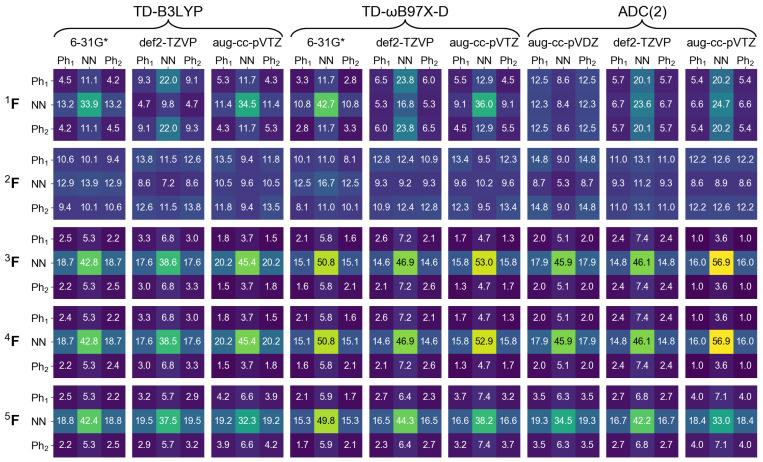
FTDM matrices ${}^{1}F,\ldots {,}^{5}F$ for the $S_{0}\to S_{1}$ transition of azobenzene calculated with nine methods indicated in the top. FTDM elements are expressed in %, i.e., multiplied by 100.

**Figure 3 molecules-26-04245-f003:**
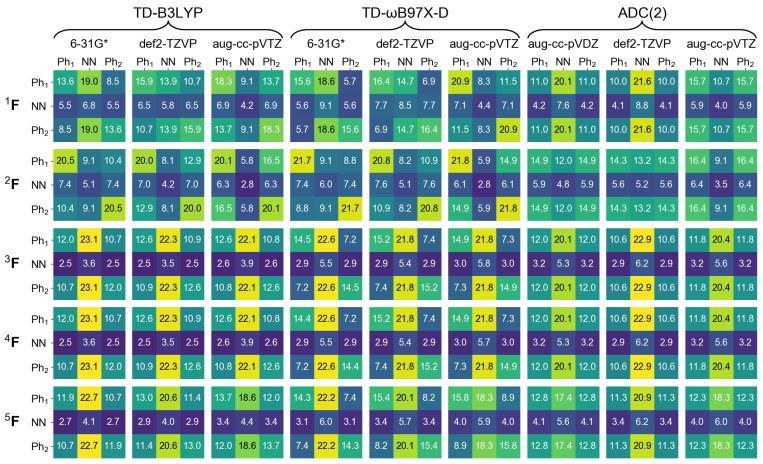
FTDM matrices ${}^{1}F,\ldots {,}^{5}F$ for the $S_{0}\to S_{1}$ transition of azobenzene calculated with nine methods indicated in the top. FTDM elements are expressed in %, i.e., multiplied by 100.

**Figure 4 molecules-26-04245-f004:**
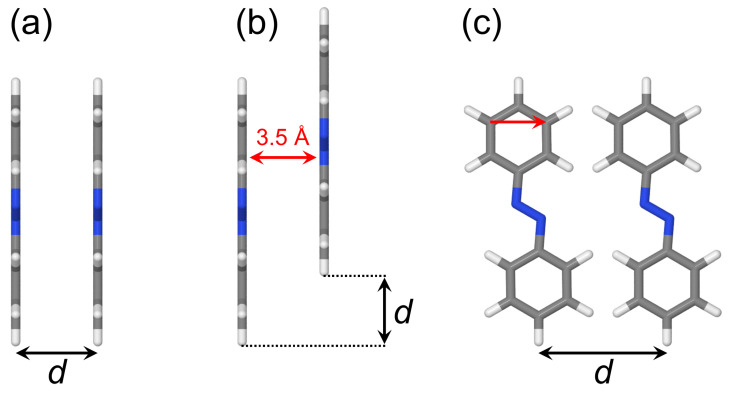
Dimer models studied in the present work. (**a**) Cofacial π-stacked dimer. (**b**) Slip-stacked dimer. (**c**) In-plane dimer. *d* denotes a coordinate varied. In (**b**), the red arrow shows the distance between molecular planes which was kept at 3.5 Å. In (**c**), the red arrow shows the translation vector calculated as a difference between coordinates of two carbon atoms connected by the arrow.

**Figure 5 molecules-26-04245-f005:**
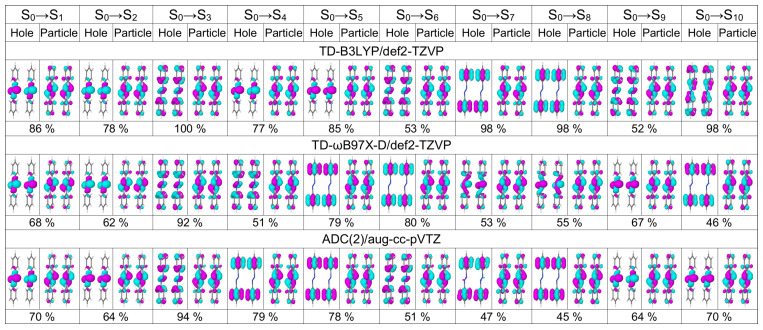
Dominant natural transition orbital pairs for the lowest ten transitions of the cofacial π-stacked dimer d=3.5 Å calculated at the TD-B3LYP/def2-TZVP (**top row**), TD-ωB97X-D/def2-TZVP (**middle row**), and ADC(2)/aug-cc-pVTZ (**bottom row**) levels of theory.

**Figure 6 molecules-26-04245-f006:**
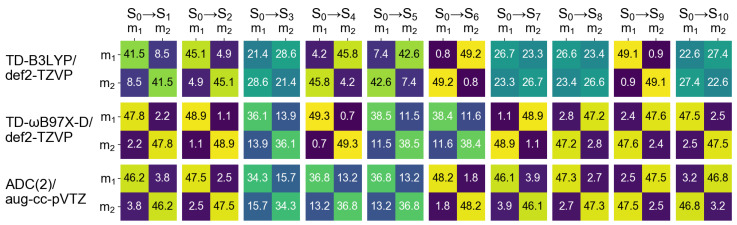
${}^{5}F$ matrices for the lowest ten transitions of the cofacial π-stacked dimer d=3.5 Å calculated at the TD-B3LYP/def2-TZVP (**top row**), TD-ωB97X-D/def2-TZVP (**middle row**), and ADC(2)/aug-cc-pVTZ (**bottom row**) levels of theory. $X,Y=m_{1},m_{2}$; m stands for “monomer”.

**Figure 7 molecules-26-04245-f007:**
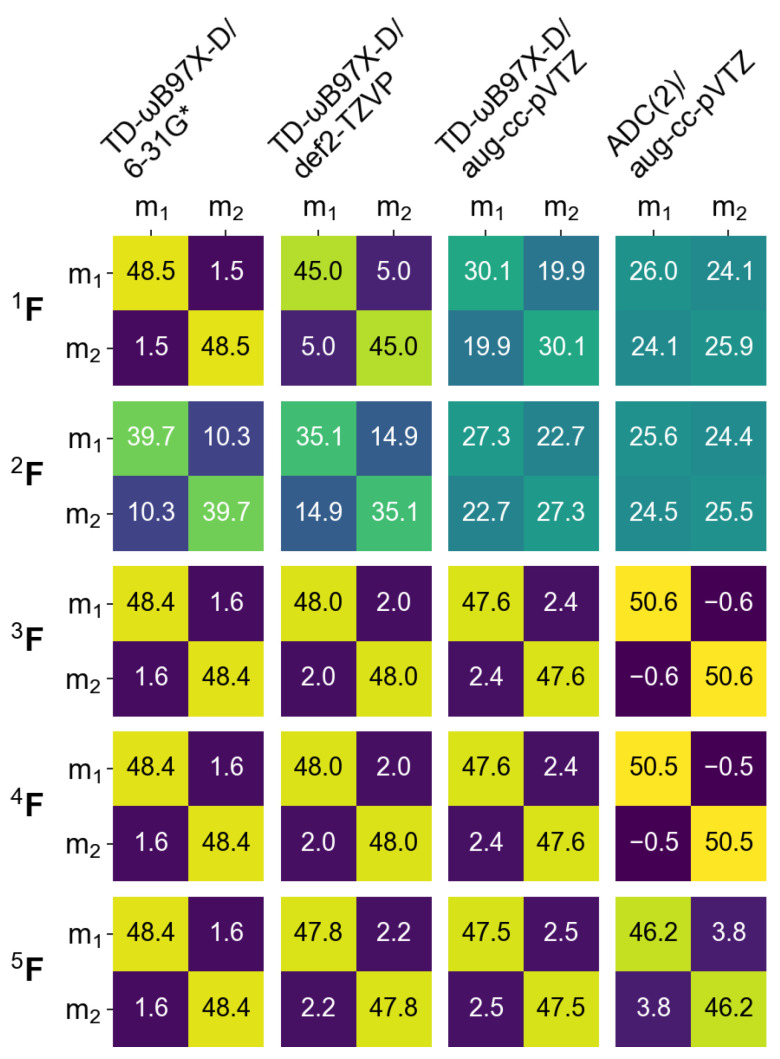
FTDM matrices ${}^{1}F,\ldots {,}^{5}F$ for the $S_{0}\to S_{1}$ transition of the cofacial π-stacked dimer d=3.5 Å calculated with four methods indicated in the top. $X,Y=m_{1},m_{2}$; m stands for “monomer”.

**Figure 8 molecules-26-04245-f008:**
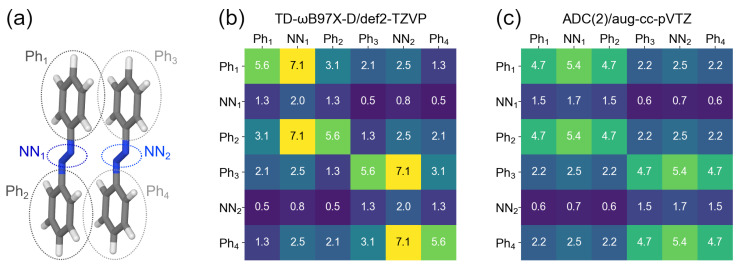
(**a**) Partitioning of the dimer d=3.5 Å to fragments. (**b**) ${}^{5}F$ matrix for the $S_{0}\to S_{3}$ transition calculated at the TD-ωB97X-D/def2-TZVP level. (**c**) ${}^{5}F$ matrix for the $S_{0}\to S_{3}$ transition calculated at the ADC(2)/aug-cc-pVTZ level.

**Figure 9 molecules-26-04245-f009:**
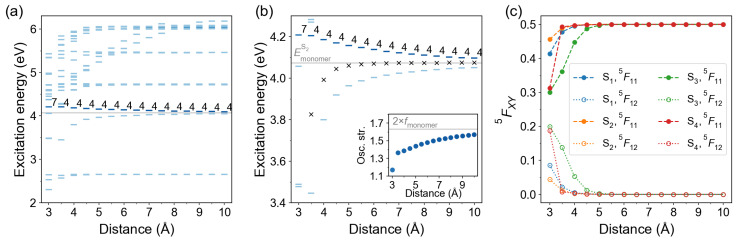
Distance dependence of excitation energies and FTDM matrix elements for π-stacked azobenzene dimers. (**a**) Excitation energies of twenty lowest transitions. The bars are colored according to the oscillator strength of the transitions, with a darker color corresponding to a larger oscillator strength. The bright state labels are printed. The light grey line marks the excitation energy of the bright monomeric transition $S_{0}\to S_{2}$. (**b**) Excitation energies of the lowest $\pi \pi^{*}$ states. Black crosses show the splitting center $\frac{E_{S_{3}}+E_{S_{4}}}{2}$. The inset shows distance dependence of the oscillator strength of the bright transition. The light grey line in the inset marks twice oscillator strength of the monomeric bright transition. (**c**) FTDM matrix elements ${}^{5}F_{11}$ and ${}^{5}F_{12}$ quantifying local and CT contributions, respectively, for the lowest four electronic transitions.

**Figure 10 molecules-26-04245-f010:**
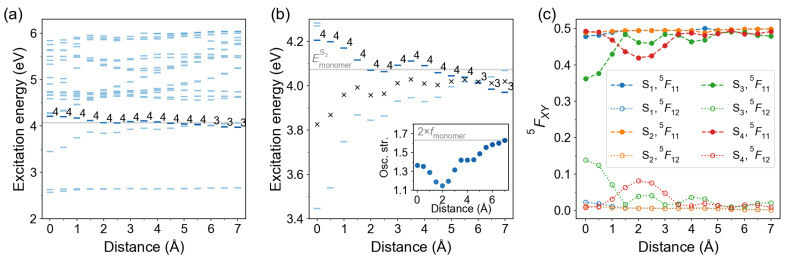
Distance dependence of excitation energies and FTDM matrix elements for slip-stacked azobenzene dimers. (**a**) Excitation energies of twenty lowest transitions. The bars are colored according to the oscillator strength of the transitions, with a darker color corresponding to a larger oscillator strength. The bright state labels are printed. The light grey line marks the excitation energy of the bright monomeric transition $S_{0}\to S_{2}$. (**b**) Excitation energies of the lowest $\pi \pi^{*}$ states. Black crosses show the splitting center $\frac{E_{S_{3}}+E_{S_{4}}}{2}$. The inset shows distance dependence of the oscillator strength of the bright transition. The light grey line in the inset marks twice oscillator strength of the monomeric bright transition. (**c**) FTDM matrix elements ${}^{5}F_{11}$ and ${}^{5}F_{12}$ quantifying local and CT contributions, respectively, for the lowest four electronic transitions.

**Figure 11 molecules-26-04245-f011:**
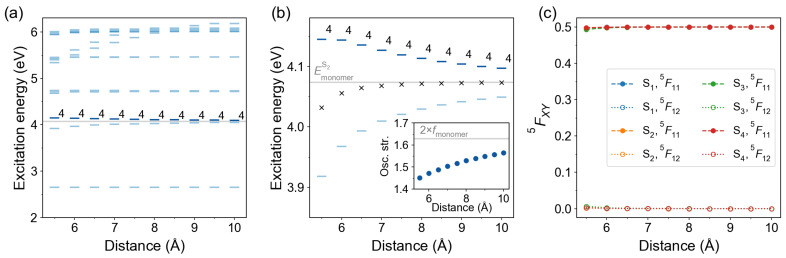
Distance dependence of excitation energies and FTDM matrix elements for in-plane azobenzene dimers. (**a**) Excitation energies of twenty lowest transitions. The bars are colored according to the oscillator strength of the transitions, with a darker color corresponding to a larger oscillator strength. The bright state labels are printed. The light grey line marks the excitation energy of the bright monomeric transition $S_{0}\to S_{2}$. (**b**) Excitation energies of the lowest $\pi \pi^{*}$ states. Black crosses show the splitting center $\frac{E_{S_{3}}+E_{S_{4}}}{2}$. The inset shows distance dependence of the oscillator strength of the bright transition. The light grey line in the inset marks twice oscillator strength of the monomeric bright transition. (**c**) FTDM matrix elements ${}^{5}F_{11}$ and ${}^{5}F_{12}$ quantifying local and CT contributions, respectively, for the lowest four electronic transitions.

**Figure 12 molecules-26-04245-f012:**
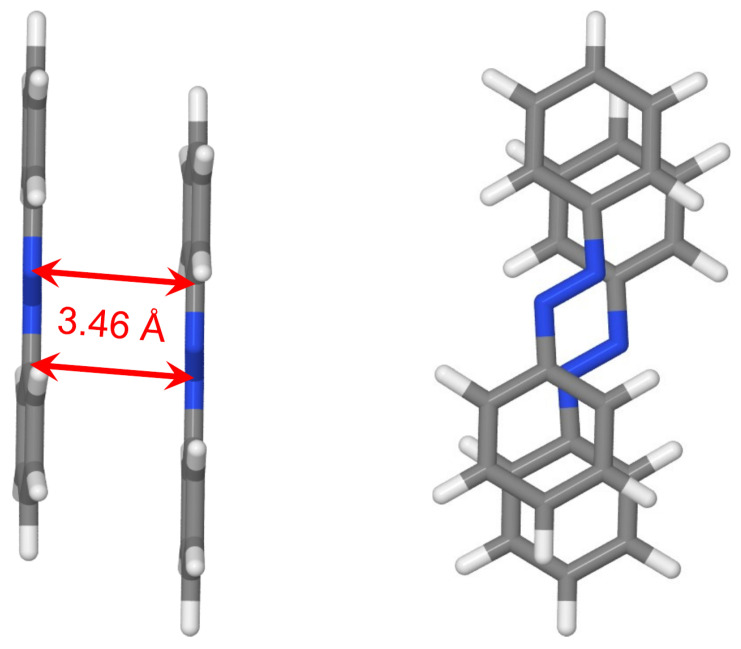
Two views of the optimized dimer geometry. The shortest intermolecular CN distances are shown. The geometry was optimized at the B3LYP+D3(BJ)/def2-TZVP level of theory.

**Figure 13 molecules-26-04245-f013:**
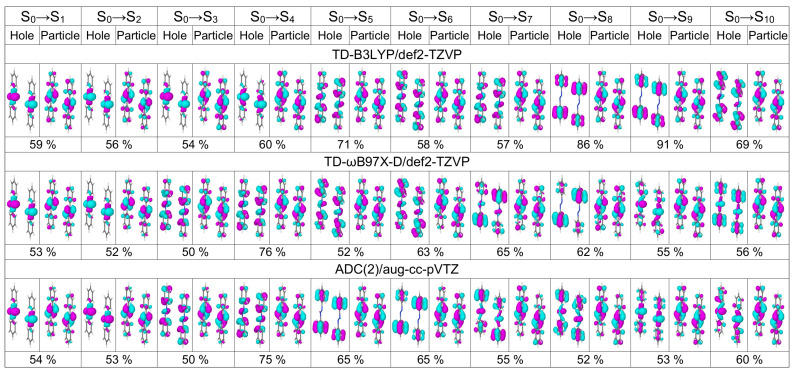
Dominant natural transition orbital pairs for the lowest ten transitions of the optimized dimer calculated at the TD-B3LYP/def2-TZVP (**top row**), TD-ωB97X-D/def2-TZVP (**middle row**), and ADC(2)/aug-cc-pVTZ (**bottom row**) levels of theory.

**Figure 14 molecules-26-04245-f014:**
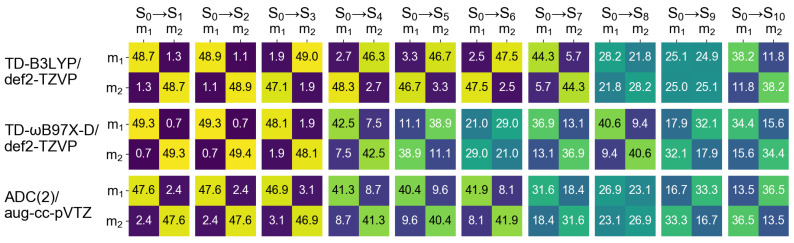
${}^{5}F$ matrices for ten excited states of the optimized dimer calculated at the TD-B3LYP/def2-TZVP (**top row**), TD-ωB97X-D/def2-TZVP (**middle row**), and ADC(2)/aug-cc-pVTZ (**bottom row**) levels of theory. $X,Y=m_{1},m_{2}$; m stands for “monomer”.

**Table 1 molecules-26-04245-t001:** Vertical excitation energies in eV and oscillator strengths (in parentheses) of the lowest five electronic transitions of *trans* azobenzene calculated with TD-B3LYP and TD-ωB97X-D and three basis sets (6-31G*, def2-TZVP, aug-cc-pVTZ) at the B3LYP/def2-TZVP and B3LYP/6-31G* optimized geometries. The brightest transitions are shown in bold.

	TD-B3LYP			TD-ωB97X-D		
	6-31G*	def2-TZVP	aug-cc-pVTZ	6-31G*	def2-TZVP	aug-cc-pVTZ
	B3LYP/def2-TZVP geometry					
$S_{0}\to S_{1}$	2.52 (0.00)	2.52 (0.00)	2.52 (0.00)	2.66 (0.00)	2.66 (0.00)	2.66 (0.00)
$S_{0}\to S_{2}$	**3.85 (0.78)**	**3.74 (0.77)**	**3.71 (0.76)**	**4.19 (0.83)**	**4.07 (0.81)**	**4.05 (0.81)**
$S_{0}\to S_{3}$	4.18 (0.00)	4.10 (0.05)	4.09 (0.05)	4.81 (0.03)	4.71 (0.03)	4.70 (0.03)
$S_{0}\to S_{4}$	4.18 (0.05)	4.10 (0.00)	4.09 (0.00)	4.82 (0.00)	4.73 (0.00)	4.71 (0.00)
$S_{0}\to S_{5}$	4.90 (0.00)	4.80 (0.00)	4.77 (0.00)	5.60 (0.00)	5.46 (0.00)	5.43 (0.00)
	B3LYP/6-31G* geometry					
$S_{0}\to S_{1}$	2.55 (0.00)	2.55 (0.00)	2.55 (0.00)	2.69 (0.00)	2.68 (0.00)	2.68 (0.00)
$S_{0}\to S_{2}$	**3.77 (0.77)**	**3.66 (0.76)**	**3.63 (0.76)**	**4.10 (0.82)**	**3.98 (0.80)**	**3.96 (0.80)**
$S_{0}\to S_{3}$	4.11 (0.00)	4.03 (0.05)	4.01 (0.05)	4.74 (0.03)	4.64 (0.03)	4.63 (0.03)
$S_{0}\to S_{4}$	4.11 (0.05)	4.03 (0.00)	4.02 (0.00)	4.75 (0.00)	4.66 (0.00)	4.64 (0.00)
$S_{0}\to S_{5}$	4.83 (0.00)	4.73 (0.00)	4.70 (0.00)	5.52 (0.00)	5.38 (0.00)	5.35 (0.00)

**Table 2 molecules-26-04245-t002:** Vertical excitation energies in eV and oscillator strengths (in parentheses) of the lowest five electronic transitions of *trans* azobenzene calculated with ADC(2) and five basis sets (aug-cc-pVDZ, aug-cc-pVTZ, aug-cc-pVQZ, def2-TZVP, def2-QZVP) at the B3LYP/def2-TZVP and B3LYP/6-31G* optimized geometries. The brightest transitions are shown in bold.

	ADC(2)				
	aug-cc-pVDZ	aug-cc-pVTZ	aug-cc-pVQZ	def2-TZVP	def2-QZVP
	B3LYP/def2-TZVP geometry				
$S_{0}\to S_{1}$	2.81 (0.00)	2.77 (0.00)	2.77 (0.00)	2.79 (0.00)	2.77 (0.00)
$S_{0}\to S_{2}$	**4.19 (0.90)**	**4.15 (0.89)**	**4.14 (0.89)**	**4.20 (0.91)**	**4.15 (0.90)**
$S_{0}\to S_{3}$	4.58 (0.03)	4.53 (0.03)	4.52 (0.03)	4.57 (0.03)	4.53 (0.03)
$S_{0}\to S_{4}$	4.59 (0.00)	4.54 (0.00)	4.53 (0.00)	4.57 (0.00)	4.54 (0.00)
$S_{0}\to S_{5}$	5.32 (0.00)	5.27 (0.00)	5.26 (0.00)	5.33 (0.00)	5.28 (0.00)
	B3LYP/6-31G* geometry				
$S_{0}\to S_{1}$	2.84 (0.00)	2.80 (0.00)	2.80 (0.00)	2.82 (0.00)	2.80 (0.00)
$S_{0}\to S_{2}$	**4.10 (0.89)**	**4.06 (0.89)**	**4.05 (0.89)**	**4.11 (0.90)**	**4.06 (0.89)**
$S_{0}\to S_{3}$	4.51 (0.03)	4.46 (0.03)	4.44 (0.03)	4.49 (0.03)	4.45 (0.03)
$S_{0}\to S_{4}$	4.52 (0.00)	4.46 (0.00)	4.45 (0.00)	4.50 (0.00)	4.46 (0.00)
$S_{0}\to S_{5}$	5.24 (0.00)	5.18 (0.00)	5.17 (0.00)	5.24 (0.00)	5.19 (0.00)

**Table 3 molecules-26-04245-t003:** Vertical excitation energies in eV and oscillator strengths (in parentheses) of the lowest ten electronic transitions of the cofacial π-stacked azobenzene dimer d=3.5 Å calculated with TD-B3LYP/def2-TZVP, TD-ωB97X-D/def2-TZVP, and ADC(2)/aug-cc-pVTZ. The dimer geometry is constructed from the B3LYP/def2-TZVP optimized monomer geometry. The brightest transitions are shown in bold.

	TD-B3LYP/def2-TZVP	TD-ωB97X-D/def2-TZVP	ADC(2)/aug-cc-pVTZ
$S_{0}\to S_{1}$	2.36 (0.00)	2.57 (0.00)	2.65 (0.00)
$S_{0}\to S_{2}$	2.46 (0.00)	2.63 (0.00)	2.72 (0.00)
$S_{0}\to S_{3}$	2.82 (0.00)	3.45 (0.00)	3.46 (0.00)
$S_{0}\to S_{4}$	3.28 (0.00)	**4.21 (1.37)**	4.04 (0.00)
$S_{0}\to S_{5}$	3.37 (0.00)	4.27 (0.00)	4.04 (0.00)
$S_{0}\to S_{6}$	3.40 (0.00)	4.28 (0.00)	**4.20 (1.50)**
$S_{0}\to S_{7}$	3.45 (0.00)	4.59 (0.00)	4.47 (0.04)
$S_{0}\to S_{8}$	3.46 (0.00)	4.66 (0.00)	4.48 (0.00)
$S_{0}\to S_{9}$	**3.88 (1.24)**	4.67 (0.00)	4.51 (0.00)
$S_{0}\to S_{10}$	4.06 (0.00)	4.73 (0.04)	4.56 (0.00)

**Table 4 molecules-26-04245-t004:** Vertical excitation energies in eV and oscillator strengths (in parentheses) of the lowest ten electronic transitions of the optimized B3LYP+D3/def2-TZVP dimer calculated with TD-B3LYP/def2-TZVP, TD-ωB97X-D/def2-TZVP, and ADC(2)/aug-cc-pVTZ. The brightest transitions are shown in bold.

	TD-B3LYP/def2-TZVP	TD-ωB97X-D/def2-TZVP	ADC(2)/aug-cc-pVTZ
$S_{0}\to S_{1}$	2.51 (0.00)	2.65 (0.00)	2.74 (0.00)
$S_{0}\to S_{2}$	2.52 (0.00)	2.65 (0.00)	2.74 (0.00)
$S_{0}\to S_{3}$	3.23 (0.02)	3.85 (0.00)	3.91 (0.00)
$S_{0}\to S_{4}$	3.23 (0.00)	**4.10 (1.15)**	**4.09 (1.24)**
$S_{0}\to S_{5}$	3.37 (0.05)	4.52 (0.00)	4.36 (0.00)
$S_{0}\to S_{6}$	3.38 (0.00)	4.56 (0.13)	4.37 (0.01)
$S_{0}\to S_{7}$	3.50 (0.00)	4.60 (0.00)	4.42 (0.02)
$S_{0}\to S_{8}$	3.82 (0.13)	4.61 (0.01)	4.42 (0.00)
$S_{0}\to S_{9}$	3.84 (0.00)	4.67 (0.08)	4.45 (0.00)
$S_{0}\to S_{10}$	**3.84 (0.82)**	4.70 (0.00)	4.47 (0.15)

## Data Availability

The data presented in this study are available in the article, Appendix A, and on request from the corresponding author.

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
