# Peer review of "On the Low-Lying Electronically Excited States of Azobenzene Dimers: Transition Density Matrix Analysis"

_molecules, 2021, doi:10.3390/molecules26144245_

Round 1

Reviewer 1 Report

  • The ms describes the results of applying the density matrix analysis to excited states of azobenzene monomer and dimers. Several types of dimers - co-facial p-stacked, slip-stacked and in-plane dimers are considered. The geometry optimized dimer structure is also a subject of the TD DFT and ADC(2) calculations. Several levels of theory and their combinations were employed. 
  • In my opinion however, the ms lacks such important part as Conclusion(s). Whereas an interested reader can squeeze his/her conclusions from section 3 (Discussion), a short conclusion from the author is absolutely needed.
  • Another point that needs addition is a complete absence of the experimental observables. There are numerous data on experimental spectra of azobenzenes in different states and media and at least some should be provided. Including such data will also make the paper more interesting to general readers.
  • Taking into account that azobenzene derivatives are currently the focus of numerous researches owing to their photochromic properties, the ms is certainly of interest and can be published after minor corrections.

Author Response

We thank the reviewer for the evaluation of our manuscript.

In my opinion however, the ms lacks such important part as Conclusion(s). Whereas an interested reader can squeeze his/her conclusions from section 3 (Discussion), a short conclusion from the author is absolutely needed.

According to the Molecules template, the Conclusions section is not mandatory. Therefore, it was not present in the original version of the manuscript. We have now added a short Conclusions section in the revised version of the manuscript.

Another point that needs addition is a complete absence of the experimental observables. There are numerous data on experimental spectra of azobenzenes in different states and media and at least some should be provided. Including such data will also make the paper more interesting to general readers.

We have added the following on Page 10 of the revised version:

At the same time, all three methods predict a blue shift of the bright transition in comparison to the monomer, as expected based on the Kasha theory [22] (compare Tables 3 and 1,2). This shift is 0.14 eV for TD-DFT and 0.05 eV for ADC(2). To make a connection to experimental studies, we note that blue shifts of about 0.1 eV have been observed upon micellization of azobenzene-functionalized surfactants [5], and blue shifts of about 0.6 eV have been found in azobenzene-containing self-assembled monolayers [3].

Reviewer 2 Report

In this article, the author compared five definitions of fraction of transition density matrix to understand the excited states of azobenzene dimers. The author concluded that the transitions to the lowest exciton states of azobenzene dimers are dominated by local excitations, while charge transfer contributions can not be neglected in several cases. 

This article is well written. It can be published in "Molecule" directly.

Author Response

We thank the reviewer for the evaluation of our manuscript.

Reviewer 3 Report

I have read with interest the paper by Titov. The author shows how different fragmentation measures of the CIS transition density matrix (TDM) behave for electronic excitations in the azobenzene monomer and dimer. The presentation is very clear and didactic, although no new method is introduced. The discussion is performed on TD-DFT and ACD(2) TDMs computed with two functionals and several basis sets, at a number of selected geometries. The authors shows clearly that the traditional natural transition orbital analysis is not useful in complex systems, in agreement with current knowledge, and that a fragment analysis of the TDM is able to easily classify the electronic transitions, isolating charge transfer excitations from local ones. Variation of the separation between the monomers in the dimer is also exploited to rationalize the nature of the transitions and to corroborate the fragment analysis. 

I have only a suggestion that, in my opinion, would contribute to improve the manuscript and a simple scientific comment.

  • I would have enjoyed to see the presentation of the Kasha model way before Section 3. It is a succinct, yet clear introduction that would allow the reader to understand the results of Section 2.
  • The 1F and 2F descriptors have no justification beyond semiempirical methods where no overlap is included. Similarly, 19-20 and 21, Mulliken and Löwdin-like, respectively, should inherit in some way the problems of these partitions as the size of the basis set increases. Given the large number of data reported, a comment on whether these well known problems in the case of population analyses are also found here is due
  • Real space partitions of the TDM, which are orbital invariant, exist since many years ago. Although I am not asking the author to compute them, a short note on this option together with a few references is also necessary. 

Author Response

We thank the reviewer for the evaluation of our manuscript.

I would have enjoyed to see the presentation of the Kasha model way before Section 3. It is a succinct, yet clear introduction that would allow the reader to understand the results of Section 2.

We have now introduced the Kasha model in the Introduction section (Section 1).

The 1F and 2F descriptors have no justification beyond semiempirical methods where no overlap is included.

We have added the following sentence on Page 5 of the revised version:

We note, however, that there is no rigorous justification for Equations (19) and (20).

Similarly, 19-20 and 21, Mulliken and Löwdin-like, respectively, should inherit in some way the problems of these partitions as the size of the basis set increases. Given the large number of data reported, a comment on whether these well known problems in the case of population analyses are also found here is due

We have added the following sentences on Page 19 of the revised version:

We also note that FTDM matrices including the AO overlap (Equations(21)(23)) are basis set dependent, similarly to Mulliken and Löwdin population analyses. And, moreover, the Mulliken-type FTDM matrices (Equations(21) and (22)) may, in principle, have unphysical negative elements (“fractions”).

Real space partitions of the TDM, which are orbital invariant, exist since many years ago. Although I am not asking the author to compute them, a short note on this option together with a few references is also necessary. 

We have added the following sentence (including two new references) on Page 19 of the revised version:

Finally, it should be noted that there are other approaches for an atom-related analysis of electronically excited states [79,80].